# [Re] Plan, Verify and Switch: Integrated Reasoning with Diverse X-of-Thoughts

## Abstract

In this work, we aim to reproduce the EMNLP 2023 paper of Liu et al. (2023) titled *Plan, Verify and Switch: Integrated Reasoning with Diverse X-of-Thoughts*. In the original paper, XoT - a prompting method which is designed to enhance LLMs' ability to deal with mathematical problems - is introduced and tested using GPT-3.5. The experiments showed that the technique outperforms the existing ones. We seek not only to ascertain the effectiveness of the method for a smaller model, the Phi-2, but also to expand the ideas of the paper by integrating metacognitive evaluation and broadening one of its modules.

## 1 Introduction

In the paper under review, the 'X-of-Thoughts' method is introduced. This is a prompting approach in which the most effective technique among CoT (Chain of Thought), PoT (Program of Thought), and EoT (Equation of Thought) is selected, implemented, and validated in an iterative process until an accurate solution is found. The objective is to significantly enhance the capability of Large Language Models (LLMs) in solving mathematical problems, addressing a key aspect of human intelligence. The issue of LLM's limited proficiency in mathematics has been a prominent focus within the research community. However, most studies have concentrated on developing a single method, such as CoT, zero-shot prompting, or ToT. This singular focus is ripe for improvement, as each method has its limitations: CoT underperforms in computation, PoT struggles with effectively handling problems involving unknown variables, and so on. An existing research strategy involves selecting the most appropriate method between CoT and PoT. However, this approach lacks the flexibility for dynamic method switching and lacks rigorous validation. The 'X-of-Thoughts' method seeks to bridge this gap, offering a more versatile and robust solution for mathematical problem-solving using LLMs. Our work aims to address both the reproduction of the original paper using Phi-2 and its extension.

## 2 Scope of reproducibility

Our work focuses on determining whether the following conclusions, which the authors drew from conducting experiments using GPT-3.5, also apply to Phi-2:

- Conclusion 1: XoT achieves improved performance in the overwhelming majority of datasets. This improvement is particularly evident in datasets that contain questions with more than three reasoning steps, as in this case, the three methods act complementarily. In contrast, in simpler datasets where each method separately exhibits good performance, this complementarity is not as pronounced.

- Conclusion 2: each of the three modules (planning, reasoning, verification) contributes to an enhanced performance.

- Conclusion 3: XoT slightly outperforms the majority-voting, while the superiority of XoT's performance increases in cases where one method is significantly more suitable for a specific problem compared to the others.

## 3  Methodology

Due to limited resources, we performed experiments on one of the 10 datasets used in the original paper - GSM8K (Cobbe et al., 2021). We chose GSM8K for 2 reasons:

- It is the second-largest dataset in the original paper.

- It covers a wide variety of grade school math problems.

After examining the code provided by the authors, we proceeded with the necessary changes for reproduction. We started by modifying the code in order to enable the execution with the Phi-2 model. Then, given that when using the original prompt for the plan module, the Phi-2 model always returned an empty string as output, we performed prompt tuning in the plan module's prompt to achieve the desired results with Phi-2. Having adapted the implementation, we conducted experiments: we divided the 1319 samples into batches of 100 due to limited resources and repeated the experiments conducted in the original paper. Finally, we added scripts for combining the results from the batches to obtain the overall outcome, and for visualizing the results.

### 3.1  Dataset

GSM8K is a dataset of 8.5K high quality linguistically diverse grade school math word problems created by human problem writers. The dataset is segmented into 7.5K training problems and 1K test problems. These problems take between 2 and 8 steps to solve, and solutions primarily involve performing a sequence of elementary calculations using basic arithmetic operations (addition, subtraction, multiplication, division) to reach the final answer. A bright middle school student should be able to solve every problem. It can be used for multi-step mathematical reasoning.

### 3.2  Model

Phi-2 (Microsoft et al., 2023) is a Transformer with 2.7 billion parameters. It was trained using the same data sources as Phi-1.5, augmented with a new data source that consists of various NLP synthetic texts and filtered websites (for safety and educational value). When assessed against benchmarks testing common sense, language understanding, and logical reasoning, Phi-2 showcased a nearly state-of-the-art performance among models with less than 13 billion parameters.

### 3.3  Experimental setup and code

For the calculation of the results and the evaluation scores, the scripts `run_gsm.sh` and `eval_gsm.sh`, which were adapted from the initial code provided by the original authors, are used.

For the evaluation of the results, the `eval_gsm.sh` runs the `analyze.py` script, which calculates the following scores:

- **Total accuracy**: the script calculates separately, for each of the three prompting methods, the percentage of the correct answers. This measures the single performance of each method, and assesses the strengths and weaknesses of the LLM.

- **Confusion matrices**: the script gives the confusion matrices for the two methods (PoT and EoT) that use passive and active verification. The confusion matrices depict the number of true positive, false positive, true negative and false negative instances, for each method and each type of verification. This assesses the accuracy of the verification module of the XoT framework.

- **Accuracy for fixed order of prompting methods**: the script calculates the accuracy scores when a fixed order of the three prompting methods is used, instead of the full XoT framework. For each fixed order two scores are calculated, one that takes into account only the passive verification,

and one that also takes into account the active verification. These scores are used to be compared with the XoT scores, and therefore they assess the importance of the plan module.

- **Majority Vote accuracy**: the script calculates the accuracy using the majority vote method. Specifically, it uses the results from all of the three methods and chooses the answer that appears the most times.

- **PEC ratio**: this score shows how many times each method is chosen in the XoT framework.

- **XoT main results**: the script calculates the overall accuracy of the XoT framework, and the average number of steps, which shows the reduction of steps and therefore reduction of cost and resources, compared to employing the majority vote method.

The necessary information for the reproduction of the results of our work is provided through the `README.md` in the folder with our code.

## 3.4 Computational requirements

We used the free resources provided in Google Colab for this reproduction, and specifically the T4 GPU. For batches of 100 samples, the running time was 3.25 hours using the Phi-2 model, for the computation of the results of all the modules (CoT, EoT, active verification of EoT, PoT, active verification of PoT, and plan). Consequently, the results for the total 1319 samples of the GSM8K test set needed 42.86 hours.

Regarding the extension with metacognitive evaluation of the CoT method, and the updated plan module which contains rating all of the three methods for their suitability to be used for each specific problem, the calculations with the DeepSeek model were more computationally expensive. The free resources of Google Colab (T4 GPU) were also used for the extension. The code for the metacognitive evaluation was executed in 'minibatches' of 10 samples. The metacognitive evaluation module needed 17 minutes for each minibatch. The updated plan module was executed in batches of 100 samples, where each batch needed approximately 1 hour. Therefore, the code for the extension needed 50.56 hours in total to be executed.

## 4 Extensions

Considering the constraints and future research directions highlighted, we proceeded to expand upon the original publication. Specifically, inspired by the work of Wang & Zhao (2023), which introduces the concept of metacognitive prompting, we integrated the methodology of metacognitive evaluation to appraise the generated outputs, and we incorporated the CoT method into the plan module, as in the original paper, the plan module is called upon to choose between the PoT and EoT methods, resulting in it never initiating with CoT.

## 4.1 Metacognitive evaluation

Before proceeding to analyze this specific idea, we provide—for the sake of completeness—the definition of metacognition: Metacognition is the process of thinking about one's own thinking and learning (Queen's University Canada).

Metacognitive evaluation constitutes a method for assessing the solutions generated by the model, based on prompts that encourage the LLM to delve deeper and reassess the accuracy of its reasoning and to identify potential errors in its solution.

With this extension, we aim to utilize a method that will be more effective primarily for CoT, since the proposed verification techniques are specially designed for PoT and EoT. After applying the method using Phi-2, which did not yield a sufficiently satisfactory result, we used a more powerful model, the DeepSeek LLM 7B Chat model (DeepSeek-AI et al., 2024).

### 4.2 Incorporation of CoT into the plan module

We incorporated the CoT method into the plan module, as in the original paper, the plan module is called upon to choose between the PoT and EoT methods, resulting in it never initiating with CoT. Given that with the use of Phi-2, among the methods CoT, PoT, EoT, the CoT method ranks second in accuracy (as shown in the 'Results' section below), the ability to choose CoT in the plan module will likely increase the accuracy of XoT.

We specifically instructed the LLM to rate the suitability of each method for solving each specific problem, with a number between 1 and 5. Then, the plan module multiplies these ratings with weights that correspond to the proportion of the different methods that the XoT framework chooses (as shown in the Figure 1), taking into account both the overall success of each method and the LLM's judgment for each specific case. This guides the XoT framework to try to solve the given problem, with the order that is specified in the updated plan module. The specific prompt that was used can be found in the Appendix in Section A.

## 5 Results

The Phi-2 model gives significantly different results than the ones presented in the paper, which were produced with the GPT-3.5 model.

### 5.1 Results reproducing the original paper

Due to using a smaller model than GPT-3.5, the results with the Phi-2 model do not completely support any of the three conclusions that were drawn in the original paper. Regarding the claim that XoT achieves improved performance in the overwhelming majority of datasets, this is not observed for every case of the fixed order of prompting methods. More specifically, the fixed order EoT, PoT, with active verification, scores significantly higher than the XoT framework.

Regarding the claim that XoT slightly outperforms the majority-voting, when using the Phi-2 model, the reverse is true, which means that the majority-voting method scores slightly higher than the XoT framework.

Finally, regarding the conclusion that each of the three modules (planning, reasoning, verification) contributes to an enhanced performance, the decreased overall accuracy of the XoT framework compared to the results of the original paper, and the higher accuracy of some of the fixed order of methods, signifies that the plan module has a significantly decreased accuracy compared to the plan module implemented with the GPT-3.5 model. The only conclusion that is qualitatively ascertained is the contribution of the active verification incorporation to the reduction of false positive rate, as shown in the confusion matrices in the Figure 2.

However, in the original paper, the authors briefly acknowledge the accuracies generated for the XoT framework with different base models, such as Llama-2 series. Given that the Phi-2 model has been proven to score similarly to Llama2-70B in several math-related tasks (Microsoft et al., 2023), and the original paper reporting an accuracy of 57.9% with the Llama-70B model in the GSM8K dataset, the XoT accuracy with the Phi-2 model is justified.

The analytical results are shown in the Table 1, in the Figure 1 and in the Figure 2.

### 5.2 Results beyond the original paper

The results with implementing the additional experiment with the metacognitive evaluation and the updated plan module are shown in the table 2 and the Figure 3.

Overall, a slight decrease in the accuracy is observed compared to the previous results. This further confirms the point that the plan module is not as effective when a smaller LLM, compared to the GPT-3.5 model, is used. If the plan module had more accurate results, given that the CoT method has the second higher accuracy score compared to the other prompting methods, the overall accuracy would have been increased.

Table 1: Results reproducing the original paper. 'Assert' means passive and active verification are taken into account. 'Passive' means only passive verification is taken into account. 'Plan' means the plan module is taken into account. The accuracy scores, with passive and active verification, that are higher than the XoT accuracy, are in bold.

| ACCURACY METRICS | Phi-2 | GPT-3.5-turbo |
|---|---|---|
| Assert PoT & EoT | **0.6005** | 0.7794 |
| Assert PoT & CoT | 0.5679 | 0.8180 |
| Assert EoT & PoT | **0.6346** | 0.8105 |
| Assert EoT & CoT | 0.4405 | 0.8279 |
| Passive PoT & CoT | 0.6641 | 0.7862 |
| Passive EoT & CoT | 0.5193 | 0.8143 |
| Assert EoT & PoT & CoT | 0.5603 | 0.8302 |
| Assert PoT & EoT & CoT | **0.5883** | 0.8294 |
| Plan & PoT & EoT | 0.5246 | 0.7293 |
| Plan & PoT & CoT | 0.4541 | 0.7983 |
| Plan Assert PE | **0.6254** | 0.7991 |
| Plan Passive XoT | 0.6322 | 0.8112 |
| XoT main results | 0.5739 | 0.8362 |
| Majority Vote | **0.5883** | 0.8241 |

Figure 1: The proportion of different methods that XoT finally chooses as the answer, for the Phi-2 (left) and GPT-3.5 (right) models.

Also, from the confusion matrices regarding the metacognitive evaluation for the CoT method, the LLM has many false positive instances compared to the active evaluation of the PoT and EoT methods, which contributes to the reduced efficiency of the overall framework.

## 6 Discussion

This reproduction was insightful regarding whether a smaller LLM could achieve similar results and whether the same conclusions could be drawn about the XoT framework. However, this reproduction did not test completely every aspect of the paper, such as testing the framework in the remaining 9 math problem

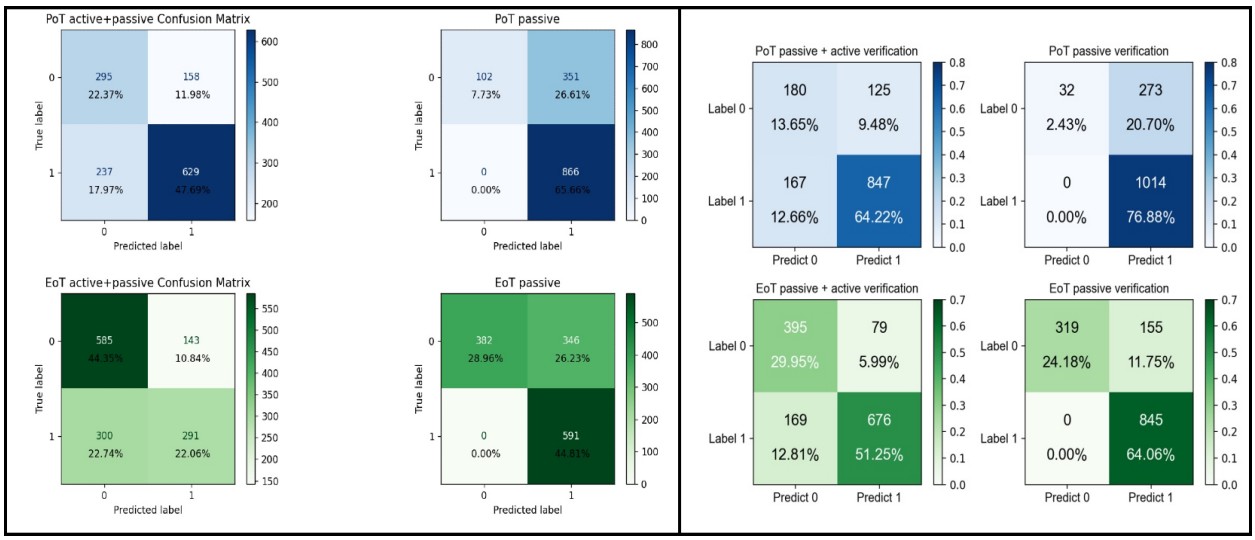

Figure 2: Confusion matrices for passive and active verification of PoT and EoT, for the Phi-2 (left) and GPT-3.5 (right) models.

Table 2: Results with adding the metacognitive evaluation and extending the plan module for all of the prompting methods.

| EXTENSIONS METRICS | Phi-2 & DeepSeek |
|---|---|
| Metacognitive evaluation accuracy | 0.6000 |
| Numeric results accuracy | 0.3594 |
| Valid correction instances | 65/844 |
| XoT main results | 0.5603 |

datasets, and testing different base models such as the Llama2 models that the original paper examined. This was mainly due to our limited computational resources.

## 6.1 What was easy

The initial code was very well written, and the original repository contains a README file that gives an overview about how to run the code in order to obtain the results and evaluation metrics. It was also really helpful that the code had the option to run the code in a subgroup of the dataset, which was helpful given that the free resources of Google Colab have a daily limit, so running the code for all the samples at once would not be possible. Additionally, it was also very helpful that the evaluation metrics were calculated by accessing the results, which were written in a jsonl file, and not during the production of the results, which meant that, for each module, we could easily combine the results of each batch into one file and then compute the metrics for the total dataset of 1319 samples. Furthermore, the authors also included the original results generated with the GPT-3.5 model, therefore the calculation of the evaluation metrics was quick and did not require running the code with the GPT-3.5 model. Most of these metrics were also included in the initial paper, but in this way, it was very straightforward to have a one-to-one comparison of the results generated by the two models.

## 6.2 What was difficult

The most challenging aspect of our reproduction was the tuning of the few-shot prompts written for each module, which was required to get the desired results using the Phi-2 model. Given that the Phi-2 model

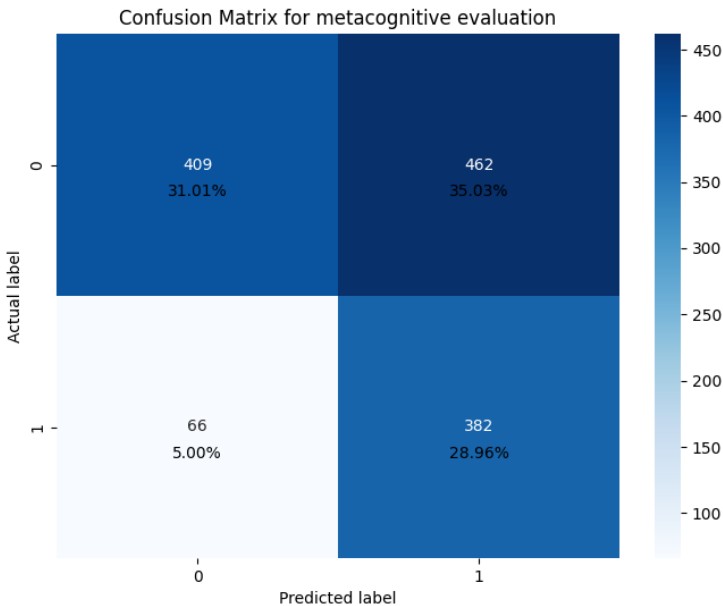

Figure 3: Confusion matrix for metacognitive evaluation of CoT reasoning.

is a completion model, whereas GPT-3.5 is a chat completion model, it is expected to work better with differently phrased prompts. With the initial prompts, the model, after solving the math problem, generated other math problems, and their solutions, until reaching the token limit. In order to retain only the solution of the math problem, we added an '<END>' token after each example in the prompt, which the LLM also generated in its response, and therefore it was very easy to retain only the desired part of the output of the model, by splitting the output string with the help of this token. Additionally, with the initial prompt of the plan module, Phi-2 generated an empty string as output for all the inputs. By phrasing the prompt differently, the Phi-2 model returned for every sample of the dataset a valid solving method. Examples showing the changes made in the prompts can be found in the Appendix in Section A.

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

## A   Appendix

We have included the prompts of each module with some example inputs and outputs.

Table 3: Example of planning module.

| |
|---|
| Input: |
| You are an expert in math reasoning. Please choose the best method to solve the math word problems between the following methods: |
| - **Python Program:** This method generates a Python program that can solve the given question. It takes in the question and possible context and produces a program. Normally, we consider using this method when the questions and contexts involve forward reasoning, such as arithmetic operations over multiple numbers, or when the questions involve complex logical operations, such as "if-else" statements. |
| - **System of linear equations:** This method builds a math model and generates a system of linear equations that contains the answer as an unknown variable. Normally, we consider using this method when the questions and contexts involve an unknown variable that must be used to build an equation, especially when the question can be better modeled with abstract mathematical declarations, or when the unknown variable appears at the beginning of the questions and needs backward reasoning to solve it. |
| Input Examples: |
| **Question**: Arnel had ten boxes of pencils with the same number of pencils in each box. He kept ten pencils and shared the remaining pencils equally with his five friends. If his friends got eight pencils each, how many pencils are in each box? |
| **Method**: System of linear equations <END> |
| **Question**: Larry spends half an hour twice a day walking and playing with his dog. He also spends a fifth of an hour every day feeding his dog. How many minutes does Larry spend on his dog each day? |
| **Method**: Python Program <END> |
| **Question**: Angela is a bike messenger in New York. She needs to deliver 8 times as many packages as meals. If she needs to deliver 27 meals and packages combined, how many meals does she deliver? |
| **Method**: System of linear equations <END> |
| **Question**: Last year Dallas was 3 times the age of his sister Darcy. Darcy is twice as old as Dexter who is 8 right now. How old is Dallas now? |
| **Method**: System of linear equations <END> |
| **Question**: A small poultry farm has 300 chickens, 200 turkeys and 80 guinea fowls. A strange, incurable disease hit the farm and every day the farmer lost 20 chickens, 8 turkeys and 5 guinea fowls. After a week, how many birds will be left in the poultry? |
| **Method**: Python Program <END> |
| **Question**: There were nine computers in the server room. Five more computers were installed each day, from monday to thursday. How many computers are now in the server room? |
| **Method**: Python Program <END> |
| **Question**: Alyssa, Keely, and Kendall ordered 100 chicken nuggets from a fast-food restaurant. Keely and Kendall each ate twice as many as Alyssa. How many did Alyssa eat? |
| **Method**: System of linear equations <END> |
| **Question**: Conor can chop 12 eggplants, 9 carrots, and 8 potatoes in a day. If he works 4 times a week, how many vegetables can he chop? |
| **Method**: Python Program <END> |
| **Question**: Cynthia eats one serving of ice cream every night. She buys cartons of ice cream with 15 servings of ice cream per carton at a cost of $4.00 per carton. After 60 days, how much will she spend on ice cream? |
| **Method**: |
| Output: |
| "Python Program" |

Table 4: Example of PoT reasoning.

| |
|---|
| Input: |
| You are an expert in math reasoning and programming. Follow the examples and write Python code to solve the math word question. Make sure that 'ans' is a number. |
| {Examples} |
| Question: A robe takes 2 bolts of blue fiber and half that much white fiber. How many bolts in total does it take? |
| # Python code, return ans, ans should be a number |
| Output: |
| `blue_fiber = 2`
`white_fiber = blue_fiber / 2`
`total_fiber = blue_fiber + white_fiber`
`ans = total_fiber` |

Table 5: Example of EoT reasoning.

| |
|---|
| Input: |
| Follow the examples and complete the equations for the question. Assume the value of the target answer as 'ans'. Make sure that 'ans' is a number. |
| {Examples} |
| Question: James decides to run 3 sprints 3 times a week. He runs 60 meters each sprint. How many total meters does he run a week? |
| System of linear equations: (Do not simplify) |
| Output: |
| % James decides to run 3 sprints 3 times a week:
`sprints_per_week = 3 * 3`
% He runs 60 meters each sprint:
`distance_per_sprint = 60`
% How many total meters does he run a week? The answer is distance_per_sprint * sprints_per_week:
`ans = distance_per_sprint * sprints_per_week` |

Table 6: Example of CoT reasoning.

| |
|---|
| Input: |
| You are an expert in math reasoning. Please solve the math word problems step by step. |
| Q: Olivia has $23. She bought five bagels for $3 each. How much money does she have left? |
| A: Olivia had 23 dollars. 5 bagels for 3 dollars each will be 5 x 3 = 15 dollars. So she has 23 - 15 dollars left. 23 - 15 is 8. The answer is 8. <END> |
| Q: Michael had 58 golf balls. On Tuesday, he lost 23 golf balls. On Wednesday, he lost 2 more. How many golf balls did he have at the end of Wednesday? |
| A: Michael started with 58 golf balls. After losing 23 on Tuesday, he had 58 - 23 = 35. After losing 2 more, he had 35 - 2 = 33 golf balls. The answer is 33. <END> |
| Q: There were nine computers in the server room. Five more computers were installed each day, from Monday to Thursday. How many computers are now in the server room? |
| A: There were originally 9 computers. For each of 4 days, 5 more computers were added. So 5 * 4 = 20 computers were added. 9 + 20 is 29. The answer is 29. <END> |
| Q: Shawn has five toys. For Christmas, he got two toys each from his mom and dad. How many toys does he have now? |
| A: Shawn started with 5 toys. If he got 2 toys each from his mom and dad, then that is 4 more toys. 5 + 4 = 9. The answer is 9. <END> |
| Q: Janet's ducks lay 16 eggs per day. She eats three for breakfast every morning and bakes muffins for her friends every day with four. She sells the remainder at the farmers' market daily for $2 per fresh duck egg. How much in dollars does she make every day at the farmers' market? |
| A: |
| Output: |
| Every day, Janet eats 3 eggs for breakfast, so she has 16 - 3 = 13 eggs left. She bakes muffins for her friends, so she has 13 - 4 = 9 eggs left. She sells the remainder at the farmers' market for $2 per egg, so she makes 9 * 2 = $18. The answer is $18. |

Table 7: Example of metacognitive evaluation.

Input:

You are an expert in math reasoning. Please evaluate the following math solution with respect to the problem stated. Ensure to follow these steps in your evaluation:

1)Relevance to Problem Conditions: First, verify if the solution comprehensively addresses all conditions and parameters outlined in the problem statement. Highlight any aspects of the problem that might have been overlooked or misinterpreted in the solution.

2)Appropriateness of Methods and Formulas: Assess whether the methods and formulas used in the solution are suitable for the type of problem. Consider if there are more efficient or accurate methods or formulas that could have been applied.

3)Accuracy of Calculations and Logical Steps: Examine each calculation and logical step for correctness. Ensure that each step is not only correct but also clearly explained and easy to follow. Point out any errors or ambiguities in the reasoning or calculations.

4)Validity of Final Answer: Finally, confirm whether the final answer is accurate and directly answers the question posed by the problem. If the answer is incorrect, provide insights into where the solution went wrong and suggest the correct approach or answer.

Include detailed explanations for each of the above steps to ensure a thorough understanding of the solution's strengths and weaknesses. Provide the answer in your final response in the following format: Evaluation: Evaluation by performing the above steps. Conclusion: Provide whether the solution is correct or incorrect. Correct solution: Provide the correct solution. Final numeric result: Provide the final numeric result of your solution.

Problem: "Ted the T-Rex was planning to bring potato salad to the dinosaur picnic. He knows that an adult dinosaur will eat 10 lbs of potato salad, and a child will eat half as much as an adult. If there will be 20 adults and 5 children at the picnic, how many pounds of potato salad does Ted need to bring to the picnic if he hopes to have enough to feed everyone?"

Solution: "An adult dinosaur will eat 10 lbs of potato salad.

A child will eat half as much as an adult, so a child will eat $10/2 = 5$ lbs of potato salad.

There will be 20 adults and 5 children, so the total number of pounds of potato salad needed is $20 + 5 = 25$ lbs.

The answer is 25 lbs.

Output:

Evaluation: Evaluation by performing the above steps.

Conclusion: The solution is incorrect.

Correct solution: Ted the T-Rex needs to bring (20 adults * 10 lbs) + (5 children * 5 lbs) = 200 lbs + 25 lbs = 225 lbs of potato salad to the picnic.

Final numeric result: Ted needs to bring 225 lbs of potato salad to the picnic to feed everyone.

Table 8: Three method plan module with rating.

| |
|---|
| Input: |
| You are an expert in math reasoning. please rate the probability of successfully solving this problem using each of the following methods: |
| - Python Program: This method generates a Python program that can solve the given question. It takes in the question and possible context and produces a program. Normally, we consider using this method when the questions and contexts involve forward reasoning, such as arithmetic operations over multiple numbers, or when the questions involve complex logical operations, such as "if-else" statements. |
| - System of linear equations: This method builds a math model and generates a system of linear equations that contains the answer as an unknown variable. Normally, we consider using this method when the questions and contexts involve an unknown variable that must be used to build an equation, especially when the question can be better modeled with abstract mathematical declarations, or when the unknown variable appears at the beginning of the questions and needs backward reasoning to solve it. |
| - Chain of Thought: This method involves breaking down the question into a series of logical steps or considerations, almost like a narrative or a conversation with oneself. It's typically used when the problem requires multi-step reasoning, where each step builds upon the previous one. The method is particularly useful for complex problems that don't lend themselves to straightforward computational or algorithmic solutions. Instead of directly arriving at the answer, the chain of thought method involves articulating each step of the thought process, providing clarity and justification for each decision or inference made along the way. This approach is ideal for problems where the reasoning path is as important as the final answer, such as in puzzles, riddles, or scenarios with multiple variables and possible interpretations. |
| The format of your answer should be the following: |
| `Python_Program_rating:{rating}` |
| `System_of_linear_equations_rating:{rating}` |
| `Chain_of_Thought_rating:{rating}` |
| Input Examples: |
| Question: Arnel had ten boxes of pencils with the same number of pencils in each box. He kept ten pencils and shared the remaining pencils equally with his five friends. If his friends got eight pencils each, how many pencils are in each box? |
| `Python_Program_rating:3` |
| `System_of_linear_equations_rating:5` |
| `Chain_of_Thought_rating:4` |
| Question: Olivia has $23. She bought five bagels for $3 each. How much money does she have left? |
| `Python_Program_rating:2` |
| `System_of_linear_equations_rating:2` |
| `Chain_of_Thought_rating:5` |
| Question: Larry spends half an hour twice a day walking and playing with his dog. He also spends a fifth of an hour every day feeding his dog. How many minutes does Larry spend on his dog each day? |
| `Python_Program_rating:5` |
| `System_of_linear_equations_rating:2` |
| `Chain_of_Thought_rating:3` |
| Question: Benny threw bologna at his balloons. He threw two pieces of bologna at each red balloon and three pieces of bologna at each yellow balloon. If Benny threw 58 pieces of bologna at a bundle of red and yellow balloons, and twenty of the balloons were red, then how many of the balloons in the bundle were yellow? |
| Output: |
| `Python_Program_rating:3` |
| `System_of_linear_equations_rating:5` |
| `Chain_of_Thought_rating:4` |

