# OpenReview forum: "[Re] Plan, Verify and Switch: Integrated Reasoning with Diverse X-of-Thoughts"
_TMLR — Rejected by TMLR_

### Review · Reviewer_xRVE · 2024-03-23

**Summary Of Contributions:**

The paper is about reproducing results of another paper called "Plan, Verify and Switch: Integrated Reasoning with Diverse X-of-Thoughts" which appeared in EMNLP 2023 and about proposing an extension to this work.

**Audience:**

No

**Broader Impact Concerns:**

I am not sure.

**Claims And Evidence:**

No

**Requested Changes:**

Major re-writing of the entire paper following all the guidelines shown above.  The paper cannot be accepted in its current form.  Also, in the abstract you should give a brief description that clearly states what your findings are.  Did you verify or did you refute the findings of the original paper?  What are the conclusions of your proposed extension of the above paper?

**Strengths And Weaknesses:**

Below I am referring to the paper

"Plan, Verify and Switch: Integrated Reasoning with Diverse X-of-Thoughts"

which appeared in EMNLP 2023 as paper [C1].


**Strengths**

**S1.** It is important that published results are reproducible.

**S2.** It is an interesting idea to investigate the extent to which conclusions hold for large language models of different sizes - according to the authors, using Phi2, the results of the EMNLP 2023 paper are less pronounced in a smaller large language model.

**Weaknesses**

**W1.** The abstract is confusing and does not clearly convey what the goal of the authors is.

**W2.** Does reproducibility of the results in a given paper warrant a new paper?  I doubt it.  I believe this is even more true, because the authors explain in their paper that they only reproduce one out of the ten different experiments presented in [C1] due to limited computational resources.

**W3.** What is even more surprising is that the paper explains that part of the reproducibility that they contribute is actually due to the fact that they are adapting the initial code that was provided for [C1].  I assume this is publicly available - though I cannot be sure.  However, if this is the case, then the authors provide as much help as it is humanly possible for their work to be reproducible.  In that sense, half of the contributions claimed in this paper are really non-contributions as [C1]'s results are reproducible.

**W4.** The paper lacks proper introduction and motivation.  Why should the reader even care about the results that the authors present?

**W5.** The paper lacks a clear section with preliminaries, notation and explanation of different technologies and terminology that the authors use throughout the paper.  Some examples:

- In the introduction, we read "ToT". What does that mean?

- In Section 3 (introductory paragraph) the authors mention Phi-2.  What is Phi-2 at that point? The authors attempt to give a description further below, in Section 3.2, but even then the description uses terms that should be defined. For example, what are even large language models? What are transformers? Even a small paragraph devoted to these notions would make the paper much better.  Not to mention that the authors try to argue about the learning process and therefore they should probably define what they mean with samples and batches and how these potentially affect learning.

- The metrics of Section 3.3 should have a clearly defined equation for each case.

-- Table 1 discusses "passive" and "active" verification.  What are these things? Where were they defined?  It is pretty much impossible to understand what is presented especially since the authors do not provide enough explanation as to how they judge qualitatively the results that they obtain.

- Near the end of the paper (7 lines before the end in page 7) the authors talk about a "token limit". What is this thing? Should this not be defined somewhere?

**W6.** There are other presentation issues.  For example, in the main part of the paper the authors should give 1-2 instances of what they are studying.  This can be done in Section 3.1.

**W7.** Some issues with expressions here and there.  For example, the introduction starts with the sentence "In the paper under review...".  There is no paper under review.  [C1] has already undergone a review, it was accepted, and it was presented in EMNLP 2023.

---

### Review · Reviewer_44ok · 2024-03-26

**Summary Of Contributions:**

This paper investigates the reproduction of results from a previous paper focused on prompting techniques to improve the ability of LLMs to deal with mathematical problems.  The goal is to reproduce the previous paper with a new language model, Phi-2, whereas the original paper focused on the GPT-3.5 model. New experiments are conducted on one of the datasets from the original paper (GSM8K) and the results do not support the same conclusions as the original paper.  This paper also explores a few extensions of the original paper, experimentally investigating their effectiveness as well.

**Audience:**

Yes

**Claims And Evidence:**

Yes

**Requested Changes:**

See the previous section for more details regarding each of these:
- (critical) Broader experimental results, in either the datasets or language models evaluated
- (critical) More details regarding the failing of Phi-2 to produce any output without prompt engineering, and more details of the prompt engineering process that was used.
- (critical) More details given regarding exactly what was done with each extension.
- (strengthening) Some explanation of the XoT method, and all of the language models that were evaluated in the original paper.

**Strengths And Weaknesses:**

Strengths - The overall goal of the article is admirable -- to replicate previous results using a new language model.  Especially as language models grow in ability and widespread useage, it is important to understand how techniques that were successful with previous language models will translate to new ones.
The writing in the paper was generally clear and succinct.

Weaknesses - In my opinion the experimental results need to be more robust and expansive for this paper to maximize its value as reproduction of previous results.  As it is, only 1 dataset from the original paper was used, and on only one new language model.  Ideally, in my opinion, a paper like this would explore the application of several new and relevant language models to all or most of the datasets from the original paper.  It is unclear from the limited experimental results whether the conclusions made from the experiments would hold more generally or if they are an artifact somehow of the specific combination of language model and dataset that were chosen.

Many of the details, specific to what was done for this paper in particular, are lacking. For example, in the beginning of section 3 the authors state that "given that when using the original prompt for the plan module, the Phi-2 model always returned an empty string as output, we performed prompt tuning in the plan module's prompt to achieve the desired results with Phi-2."  I find this sentence to be remarkable and raise questions for several reasons.  First, why did Phi-2 not produce any output given the original prompt?  This seems quite strange and my curiosity demands further exploration and explanation of why this would occur.  Second, how was the prompt tuning performed?  What specifically was changed about the prompt?  Was there a prompt tuning methodology that was followed, or was the prompt changed in an ad-hoc manner?  If it is found to be a common occurance that prompts designed for one language model fail so completely with another, this would appear to be an important topic to enable future researchers to adapt previous prompt strategies to new language models in a principled manner.  Third, what were the desired results?  How did the authors determine that the new engineered prompt was performing well with Phi-2? This is mentioned again in the beginning of section 6.2.

Another example of a lack of detail and explanation can be found in the description of the extensions.  The metacognitive evaluation was never completely described.  How exactly does it work?  In section 4.2, when describing the incorporation of CoT into the plan module, how was the prompt (mentioned in the second paragraph of that section) determined?

In section 5.1 it appears that the original paper authors did evaluated the XoT framework with other base models.  This fact, in my opinion, should have been included earlier in the paper, for example in the introduction, as that section led the reader to believe that the only model evaluated in the previous work was GPT-3.5.

Finally, while I have not seen many reproduction papers prior to this, I feel that more explanation of the XoT method, which is the focus of the paper, would be helpful to the reader.

---

> ### Author Response · Authors · 2024-04-11
> **Re:Review of Paper2187 by Reviewer 44ok**
>
> Thank you for the constructive feedback; it was particularly helpful. As it is not possible to provide experimental results for all the datasets used in the original paper due to limited resources, we would appreciate your advice on whether providing results for 3 more datasets (making it 4 out of the 10 datasets in total) would be sufficient.

---

> > ### Comment · Reviewer_44ok · 2024-04-12
> >
> > That would definitely strengthen the paper and help with that dimension of the experimental evaluation, but if that was the only addition I personally would not consider it sufficient in and of itself.

---

### Review · Reviewer_TsEB · 2024-04-10

**Summary Of Contributions:**

This paper aims to reproduce the study presented in "Plan, Verify and Switch: Integrated Reasoning with Diverse X-of-Thoughts" by Liu et al. (2023). The original study introduces the X-of-Thought (XoT) method, which flexibly selects the most suitable reasoning procedure for a given problem. The authors revisit three claims: 1) XoT achieves improved performance across the majority of datasets; 2) each of the three modules—planning, reasoning, and verification—contributes to enhanced performance; and 3) XoT marginally outperforms majority voting. However, experiments show that these claims do not hold for Phi-2, a Large Language Model (LLM) smaller than the GPT-3.5 used in the original study. The method proposes a meta-cognition evaluation to further assess the method.

**Audience:**

No

**Claims And Evidence:**

No

**Requested Changes:**

I think the paper is fundamentally weak. Adding small experiments or revising the writing will not sway my opinion. I encourage the authors to conduct more research to derive more significant contributions.

**Strengths And Weaknesses:**

**Strength**:
- The reproduction is useful for understanding performance of the studied prompting method on a small language model.

**Weaknesses**:
- The technical contribution is marginal. The paper merely reimplements a previous paper without proposing any improvements.
- Experiments are conducted on only a single model (phi-2).
- The finding is not too surprising. The authors do not provide explanations for the observed phenomenon.
- The writing does not meet the quality of a journal publication. Honestly, the paper reads like a class project report. The introduction is too short. There is no related work section. The description of metacognition evaluation, which is one of the main contributions, is extremely vague.

---

### Decision · Action_Editor_n8of · 2024-06-02

**Recommendation:** Reject

**Comment:**

The reviewers have pointed out weakness in terms of what's missing for this to be considered full reproducibility study and I encourage authors to revise the paper accordingly. Please note that the rejection decision did not take into account reviewer's criticism of "lack of novelty" as reproducibility studies are not expected to contribute new technical ideas.

**Audience:**

TMLR solicits papers focused on reproducibility and given the interest in the original EMNLP paper, a reproducibility study would be of interest to TMLR audience.

**Claims And Evidence:**

The paper focuses on reproducibility of a previously published paper in EMNLP. The reviewers felt that the paper did not run a full reproducibility study and therefore, does not provide sufficient evidence.

**Resubmission Of Major Revision:**

The authors may consider submitting a major revision at a later time.